Differential use of salmon by vertebrate consumers: implications for conservation

Levi Taal 1 taal.levi@oregonstate.edu
Wheat Rachel E. 2
Allen Jennifer M. 1
Wilmers Christopher C. 2
1 Department of Fisheries and Wildlife, Oregon State University , Corvallis, OR , USA
2 Center for Integrated Spatial Research, Department of Environmental Studies, University of California , Santa Cruz, CA , USA
Stanford Jack
Electronic publication date: 2015 Aug 4
Publication date: 2015
Volume: 3
Electronic Location ID: e1157
Received 2015 Mar 8; Accepted 2015 Jul 15
Copyright: © 2015 Levi et al.
Copyright year: 2015
Copyright holder: Levi et al.
License: This is an open access article distributed under the terms of the Creative Commons Attribution License, which permits unrestricted use, distribution, reproduction and adaptation in any medium and for any purpose provided that it is properly attributed. For attribution, the original author(s), title, publication source (PeerJ) and either DOI or URL of the article must be cited.
License URL: https://creativecommons.org/licenses/by/4.0/

Keywords: Bear, Anadromous fish, Bald eagle, Scavenger, Marine-derived nutrients, Resource pulse

Funding: National Science Foundation GRF National Science Foundation PRFB The work was funded by National Science Foundation GRF fellowships to Taal Levi and Rachel Wheat, and an NSF PRFB fellowship to Taal Levi. The funders had no role in study design, data collection and analysis, decision to publish, or preparation of the manuscript.

==============================
Salmon and other anadromous fish are consumed by vertebrates with distinct life history strategies to capitalize on this ephemeral pulse of resource availability. Depending on the timing of salmon arrival, this resource may be in surplus to the needs of vertebrate consumers if, for instance, their populations are limited by food availability during other times of year. However, the life history of some consumers enables more efficient exploitation of these ephemeral resources. Bears can deposit fat and then hibernate to avoid winter food scarcity, and highly mobile consumers such as eagles, gulls, and other birds can migrate to access asynchronous pulses of salmon availability. We used camera traps on pink, chum, and sockeye salmon spawning grounds with various run times and stream morphologies, and on individual salmon carcasses, to discern potentially different use patterns among consumers. Wildlife use of salmon was highly heterogeneous. Ravens were the only avian consumer that fed heavily on pink salmon in small streams. Eagles and gulls did not feed on early pink salmon runs in streams, and only moderately at early sockeye runs, but were the dominant consumers at late chum salmon runs, particularly on expansive river flats. Brown bears used all salmon resources far more than other terrestrial vertebrates. Notably, black bears were not observed on salmon spawning grounds despite being the most frequently observed vertebrate on roads and trails. From a conservation and management perspective, all salmon species and stream morphologies are used extensively by bears, but salmon spawning late in the year are disproportionately important to eagles and other highly mobile species that are seasonally limited by winter food availability.

Introduction

The annual return of anadromous salmon contributes pulses of marine energy and nutrients to freshwater and terrestrial systems that propagate through food-webs and influence primary producers, invertebrates, fish, and wildlife (Willson & Halupka, 1995). Terrestrial vertebrate consumers of adult wild salmon include bears, wolves, marten, mink, and coyote, and a diverse array of avian scavengers including bald eagles, ravens, jays, mergansers, gulls, and even owls (this study; Shardlow & Hyatt, 2013). Life history variation in salmon spawning phenology can extend this resource subsidy through time for more mobile consumers that move among runs that peak at different times (Schindler et al., 2013). Consumers with distinct life histories might also use salmon differently depending on the phenology of the run and the spatial dispersion of the salmon resource. Moreover, understanding how the spatiotemporal distribution of salmon can influence multiple vertebrate consumers would increase understanding about how the needs of wildlife can be incorporated into salmon management (Levi et al., 2012).

Recent research proposed incorporating brown bear fitness into salmon management decisions (Levi et al., 2012) because their body mass, litter size, and population density are closely linked to salmon consumption (Hilderbrand et al., 1999). Bald eagles (Haliaeetus leucocephalus) might also be a promising focal taxa for incorporating wildlife needs into salmon management because they (1) are large-bodied birds and thus have higher caloric requirements than smaller bird species, (2) are primarily fish-eaters, (3) are of conservation interest to the public and tourism operators, and (4) reach much higher population densities in salmon-fed systems. However, due to their distinct life histories, bears and eagles may respond differently to spatiotemporal patterns of salmon availability.

Bears avoid winter food limitation by storing fat during the pulse of returning adult salmon. Female brown bears (Ursus arctos) nearly double their body mass as they deposit fat during the months of salmon availability in preparation for hibernation and lactation (Kingsley, Nagy & Russell, 1983). Even with reduced salmon biomass entering rivers due to commercial fishing, salmon represent roughly 60–80% of bear diets in many coastal salmon systems (Mowat & Heard, 2006). In contrast, bald eagles capitalize on resource pulses by moving long distances both locally and regionally to track asynchronous resource availability (Elliott et al., 2011). When salmon are regionally abundant in summer and early fall, salmon are in surplus to the energetic needs of eagles. In contrast to bears, eagles cannot hibernate and are limited by food availability in late winter after salmon have disappeared from the landscape (Elliott et al., 2011). This leads eagles to forage over large areas and then to congregate in the thousands on late salmon runs when salmon becomes limiting. Thus the activity of eagles and other migratory avian scavengers is expected to increasingly concentrate on late salmon runs.

The spatiotemporal dispersion of salmon resources is in part a consequence of salmon species richness, as salmonid species and/or populations have varying phenologies and prefer different spawning habitats (See Study System below). Abiotic factors such as stream morphology and habitat type might also mediate which consumers access particular salmon resources (Quinn et al., 2001). For example, forest specialists (e.g., martens) might restrict their use of salmon to small and forested streams, while large avian consumers might avoid these small streams in preference for large open areas with better escape terrain.

Further, bear activity at spawning grounds can influence availability of salmon carcasses to other consumers. After capturing salmon in rivers and streams, brown bears move carcasses to land to feed, often feeding selectively on energy-rich body parts when salmon are abundant and easy to catch (Gende, Quinn & Willson, 2001). Some estimates suggest that bears consume as little as 25% of the salmon they kill (Quinn, 2005), leaving partially-consumed carcasses available to a wide range of scavengers.

Here we report the findings of a study in which we used remotely-triggered camera traps to quantify how wildlife foraging activity varies at salmon spawning grounds (Shardlow & Hyatt, 2013), and rates of scavenging on individual salmon carcasses deposited on the forest floor. We monitored runs with distinct run timing and variable stream morphologies including small creeks in forested areas, lake shores, pools off of larger rivers (i.e., alcoves), and river flats around braided mainstems of larger rivers. We also used camera traps on roads and trails before spawning to estimate the relative abundance of carnivores as a comparison to carnivore activity on salmon spawning grounds.

Study area

Spawning ground monitoring was conducted in the Chilkat and Chilkoot drainages near Haines, Alaska (Fig. 1) from June to November 2011. Monitoring of individual carcasses took place between August and October in both 2012 and 2013. The Chilkoot River flows less than 1 km from Chilkoot Lake before reaching the ocean. Chilkoot Lake is a glacially turbid lake, approximately 6 km long and 2 km wide. Primarily sockeye (Oncorhynchus nerka), but also coho (Oncorhynchus kisutch), spawn on the shores of the lake, and pink salmon (Oncorhynchus gorbuscha) spawn in the lower river and lake. Sockeye and coho also spawn in the river upstream of the lake.

Figure 1 Study area.

(A) Chilkat and Chilkoot river systems near Haines, Alaska and locations of pink, sockeye, and chum spawning sites that we monitored with camera traps. Pink salmon are primarily available on the spawning grounds in August, but begin in late July and extend into September. Chum are available in late September and October on the Klehini and at Herman Creek, but persist into December at Chilkat Flats. Sockeye salmon are available on the spawning grounds from late July through September, but last through October on the upper Chilkat River. (B) Location of study area in Northern Southeast Alaska at the end of the Lynn Canal.

The Chilkat drainage is a larger river system with multiple tributaries. The river is braided through a wide valley and meets the Tsirku and Klehini rivers 21 miles from the ocean at an area known as the Council Grounds. Ground water wells up from an alluvial fan at the intersection of these rivers, which prevents the river from freezing. A late chum salmon (Oncorhynchus keta) run spawns in the Chilkat River near the Council Grounds from October to December and in the lower Klehini River earlier in September–October. Pink salmon spawn in creeks 10 miles and 18 miles from Haines and in parts of the upper Chilkat River from August to early September. Sockeye salmon spawn in pools in the upper Chilkat River, Mosquito Lake, and in Chilkat Lake with early and late runs spanning June through October. Coho spawn in lakes and streams throughout the watershed but not in dense aggregations. Coho salmon persist in small streams into January. We identified all run timings through direct observation while visiting salmon spawning areas identified by Alaska Department of Fish and Game. The availability of chum and coho salmon late in the year draws a large congregation of bald eagles to the Chilkat River. Black-billed magpies (Pica hudsonia), common ravens (Corvus corax), mew gulls (Larus canus), and glaucous-winged gulls (Larus glaucescens) are also abundant and feed alongside the eagles.

Methods

Spawning ground monitoring

We used twenty motion-activated infrared cameras (Bushnell Trophycam) to monitor wildlife activity on pink, chum, and sockeye spawning grounds from August to November 2011 (Figs. 1 and 2). Motion-activated cameras on spawning areas may produce biased results because larger-bodied animals can trigger the camera at a greater distance in the viewshed. We elected not to sample on a systematic interval because such sampling is likely to miss infrequent visitors unless the sampling frequency is very high (i.e., nearly continuous monitoring), which would produce an intractably large number of photos for a study of this scale. We were not concerned about any bias introduced by motion-activated cameras because this research was designed to understand qualitative and broad-scale natural history patterns.

Figure 2 Spawning ground monitoring.

Examples of recorded images of wildlife visitation to salmon spawning areas at (A–B) pools, (C–D) river flats, (E–F) small streams, (G–H) and lakeshores.

We classified stream morphology into creeks, pools, flats, and lakes. Creeks are small streams with forest cover, which includes the pink salmon spawning grounds at 10 mi and 18 mi Creek, and chum salmon spawning grounds at Herman Creek. Pools are slow moving shallow offshoots of the Chilkat River (i.e., alcoves), including Bear Flats and Mule Meadows. Flats are expansive seasonally-flooded areas of braided river with no forest cover on the Chilkat and Klehini Rivers. The lake category included the lakeshore spawners of Chilkoot Lake and Mosquito Lake (Fig. 1). We monitored dirt roads and trails on the Kelsall road system, which parallels the upper Chilkat River and branches to follow the Kelsall River and Nataga Creek tributaries (Fig. 1). We placed camera traps on roads and trails in early summer to provide an index of relative abundance of terrestrial carnivores such as black bears, brown bears, coyotes, lynx, and wolves. This method may introduce biases if some carnivores select for or against roads and trails, but encounter rates serve as a useful index of abundance. All cameras were set to take three pictures when triggered with a three second delay between successive triggers. To avoid overestimating visitation rates when the same individual or group foraged in front of the camera for an extended period, we post-processed the camera data to identify unique visitations. We defined unique group visitations (where “unique” refers to a unique encounter rather than a unique individual) as visits with a greater than two minute delay between the last photograph from one visit to the first photograph of the next. We weighted each unique group visit by the observed group size to estimate the number of individual encounters (unique visits weighted by group size per camera-day). We chose a short two-minute delay because we often observed one group of animals replace another in quick succession (e.g., subdominant sow with cubs replaced by another sow).

Salmon carcass monitoring

We used motion-activated infrared cameras (Bushnell Trophycam) to monitor wildlife visitations to, and feeding activity on, individual salmon carcasses at two mid-season sockeye runs (Chilkoot Lake and Mule Meadows, August and September) and one late season chum salmon run (Herman Creek, September and October) in both 2012 and 2013. At each site we erected a grid of eight cameras, four cameras 15 m from spawning grounds and four cameras 50 m from spawning grounds, with 150 m lateral spacing between cameras. Each camera was baited with a single salmon carcass staked to the ground. Salmon carcasses were collected from the adjacent spawning grounds and were monitoring during the period of the salmon run. All cameras were set to take three pictures when triggered with a one-minute delay between successive triggers. Cameras were checked weekly and carcasses were replaced if missing or decomposed. We defined unique visitations as visits with a greater than five minute delay between the last photograph from one visit and the first photograph of the next on any one camera at the site.

Results and Discussion

We obtained over 35,000 images of animals from salmon spawning grounds during 788 camera-days in 2011, and over 25,000 images from individual salmon carcasses during 2012 and 2013. More than 15,000 images of animals were recorded in 2012 over 675 camera-days and just over 10,000 images of animals were obtained in 2013 across 714 camera-days. There was substantial variation in wildlife activity across spawning grounds with different stream morphologies and run timing (Figs. 3 and 6).

Figure 3 Spawning ground encounter rates.

Mean individual encounter rate of salmon consumers at nine salmon spawning grounds in 2011. Sites are labeled to indicate early pink salmon runs in creeks, sockeye runs in pools and lakes (that spawn over an extended period from early to late), late chum salmon runs at flats and creeks, and the very late chum salmon run at Chilkat Flats. To account for instances where multiple individuals of the same species were recorded in one frame (e.g., flocks of gulls, sows with cubs) individual encounters are the number of unique group encounters (>2 min apart) weighted by the mean number of individuals in each group. Error bars indicate standard errors of the mean across cameras within each site.

Spawning ground monitoring, 2011

Brown bears foraged extensively on all salmon runs regardless of species, run timing, and stream morphology (Figs. 2–4), including lakeshore spawners. Ravens, which are resident forest birds, were the only birds to feed extensively on early runs of pink salmon, which were also consumed by mustelids (mink and marten) and coyotes (Fig. 4A), but ravens were not observed at 10 mile creek despite an abundance of carrion (Fig. 3). Ravens also generally fed more than other avian scavengers on sockeye spawning grounds (Fig. 4A), but they were outnumbered at Chilkoot Lake, where eagles were more often observed, and Bear Flats, where magpies were observed slightly more frequently (Fig. 3). We speculate that ravens selected against the relatively coastal spawning areas at Chilkoot Lake and 10 mile creek in favor of concurrent inland spawning areas.

Figure 4 Spawning ground encounter rates grouped by species and stream morphology.

Individual encounter rate of salmon consumers on (A) pink, sockeye, and chum salmon spawning grounds, and (B) on spawning grounds at creeks, pools, flats, and lakes during spawning ground monitoring in 2011.

Migratory avian scavengers, including eagles, gulls, and magpies did not feed on early pink salmon runs (Figs. 3 and 4A). This was likely due to a combination of early run timing and avoidance of small streams when salmon are available elsewhere at sites with more suitable stream morphology. A concurrent larger pink salmon run occurs on the more expansive Chilkoot River where large numbers of gulls and eagles feed on pink salmon, but we did not monitor this run with cameras because the river is heavily used by humans. The absence of birds, other than ravens, at early pink salmon runs was followed by a high concentration of eagle and gull activity at late chum salmon runs (Figs. 3 and 4). However, avian scavengers fed on salmon less often than brown bears at the late chum salmon run at Herman Creek. We speculate that stream morphology and run timing both influenced where these consumers preferentially fed on salmon (Figs. 3 and 4). This was particularly evident by the very few gulls observed at Herman Creek relative to Klehini Flats and Chilkat Flats.

Brown bears were observed disproportionately more on salmon spawning grounds than expected by their abundance. Black bears were the most commonly observed animal on roads and trails during summer 2011, followed by coyotes, brown bears, wolves and lynx (Fig. 5). Black bears, which are dominant salmon consumers in many systems without brown bears, were notably absent on salmon spawning grounds (Fortin et al., 2007) (Fig. 5), which suggests that risk associated with interference competition by brown bears is strong enough to prevent black bear consumption of this energetically profitable resource. Similarly, coyotes were much more abundant than wolves, but wolves were observed as frequently as coyotes on salmon spawning grounds. Lynx were not observed on spawning grounds (Fig. 5).

Figure 5 Road and trail encounter rates.

Relative abundance of terrestrial carnivores as measured by camera trapping rates on dirt roads and trails in the upper Chilkat River watershed from June to August and on salmon spawning grounds from August to October 2011. Black bears were the most commonly observed species on roads and trails but completely avoided spawning areas, which were dominated by brown bears. We observed 83 unique encounters of black bears, 42 coyotes, and 21 brown bears, 4 wolves, and 3 lynx.

Figure 6 Salmon carcass monitoring.

Recorded images of wildlife consumption of individual salmon carcasses at baited camera trap stations during 2012 and 2013. A wide variety of species were observed feeding on salmon carcasses, including brown bears (A, B), eagles, ravens, crows (C), mink, marten (D), coyotes (E), and wolves (F).

Salmon carcass monitoring, 2012–2013

A wide variety of species were observed both visiting and consuming salmon carcasses in both 2012 and 2013 (Fig. 6). Encounter rates of species varied among sites and between years (Fig. 7). We observed no differences between visitations at carcasses near to (15 m) and farther from (50 m) spawning grounds for any species (all p > 0.2). As with spawning ground monitoring in 2011, brown bears were the dominant visitors to all carcasses regardless of run timing. Although avian activity overall was observed to be higher at late runs at Herman Creek than mid-season runs at Chilkoot Lake and Mule Meadows, avian encounters were much lower on individual salmon carcasses in 2012 and 2013 than on spawning grounds in 2011. As salmon carcass sites were either 15 or 50 m from spawning grounds and often under forest cover, this could suggest that avian scavenging on partially-consumed salmon carcasses deposited by bears might be limited to larger, open areas immediately adjacent to spawning grounds or in sparsely or unforested habitats like river flats.

Figure 7 Salmon carcass encounter rate.

Mean individual encounter rate of salmon consumers visiting individual salmon carcasses at two mid-season (Chilkoot Lake and Mule Meadows) and one late season (Herman Creek) salmon runs in 2012 and 2013. Error bars indicate standard errors of the mean across cameras within each site.

Visitation of wildlife to salmon carcasses occurred consistently at each site, but consumption of carcasses was observed less frequently. While small carnivores such as mink and marten and avian scavengers were observed feeding on individual carcasses at all sites throughout the duration of the carcass monitoring, bears rarely fed on carcasses until the end of the late chum salmon run at Herman Creek. Since salmon carcasses were used to bait camera stations rather than live salmon, this could indicate a preference for predation over scavenging, or for consumption of fresher fish until the point at which salmon becomes limiting. The observation that bears frequently return to salmon carcasses after initially high grading on the most calorie-rich portion of the fish suggests that estimates of salmon consumption by bears are likely biased low (Gende, Quinn & Willson, 2001).

The extensive use of all salmon resources by brown bears suggests that bears are benefiting from a diversity of run timing by moving between asynchronous spawning aggregations to maximize their nutritional intake (Schindler et al., 2013). It is possible that relaxing harvests on lower value salmon species such as pink salmon (2014 Alaska ex-vessel value of $0.28/lb) and chum ($0.64/lb), while fishing higher value species such as king ($4.27/lb), and sockeye ($1.75/lb) for maximum sustainable yield would be an effective strategy to increase bear population productivity with less economic impact. This seems particularly plausible because pink and chum spawn early and late in the season, respectively, which extends the temporal availability of salmon biomass. However, these species typically do not permeate as far into watersheds as other salmon species, restricting access to more interior salmon predators and scavengers. It is also important to note that migration timing, when salmon are available for commercial harvest, and timing of spawning, when salmon are available to terrestrial consumers, do not always closely correspond (Boatright, Quinn & Hilborn, 2004; Doctor et al., 2010), so a focus on early or midseason runs for commercial harvest might not result in increased escapement of late spawners.

The complete absence of eagles and other avian scavengers on early pink salmon runs suggests that regional salmon availability far exceeds their energy requirements during this time of year. These avian scavengers congregate in enormous numbers at late chum salmon runs when salmon become a limiting resource (Elliott et al., 2011; ∼3,500 observed at our field site by aerial counts). The Chilkat River eagle gathering from October to January is the earliest of these congregations and the farthest north. The Squamish and Harrison river systems in southern British Columbia also support large eagle congregations that peak in January. Other much smaller congregations continue in south-coastal British Columbia and Northern Washington State though April. These few late chum salmon runs are disproportionately important resources for the larger panmictic eagle population in the Pacific Northwest, Alaska, and Western Canada (Elliott et al., 2011).

An obvious question is why other terrestrial vertebrates use salmon so much less than brown bears and avian scavengers. Bears maximize their annual energy balance by depositing fat during salmon abundance and hibernating during food scarcity, and migratory avian scavengers can track asynchronous pulses of salmon on vast spatial scales. We speculate that an inability to employ these life history strategies prevents other salmon consumers from reaching the atypical population densities (i.e., relative to elsewhere in their range) reached by bears and mobile avian scavengers in the presence of salmon. Winter food availability rather than salmon may limit the populations of most terrestrial carnivores, but the importance of salmon to each species is likely to be idiosyncratic. Wolves, for example, are more successful at hunting ungulate prey in winter than during summer, such that salmon can lead to apparent competition by maintaining wolves at high density even when ungulate biomass is low (Adams et al., 2010). Black bears do possess the hibernation strategy to effectively exploit pulsed salmon resources, and do so in many river systems without brown bears, but interference competition with brown bears prevents salmon consumption at our field site. It is unclear to what degree interspecific competition affects salmon consumption by other species, but is plausible that coyotes, which are locally abundant (Fig. 4), were observed infrequently due to interference competition from wolves. For small-bodied scavengers, such as mink and marten, the spatial distribution of salmon, rather than biomass per se, may be a more important driver of population productivity because even relatively small quantities of salmon may support the low population density of resident individuals. However, this depends on the accessibility of salmon carcasses, which can be quickly flushed from systems in the absence of large woody debris (Cederholm et al., 1999), or deposition of partially consumed carcasses by bears (Helfield & Naiman, 2006). Additionally, small carnivores may focus more on salmon carcasses when small mammals, their alternative prey, are rare, because rodents have much higher energy densities than do spawned out salmon carcasses (∼7–8 kJ/g for rodents compared to ∼3 kJ/g for spawned out salmon Cox & Secor, 2007; Hendry & Berg, 1999).

Understanding how salmon use is influenced by consumer life history, run timing and stream morphology is an important first step toward integrating wildlife needs into salmon management as part of an ongoing paradigmatic shift toward ecosystem-based fisheries management (Levin et al., 2009). Although there are quantitative differences in the nutritional quality of salmon and their accessibility, all salmon on the landscape were heavily used by bears. In contrast, the activity pattern of bald eagles suggests that fisheries management that considers their nutritional requirements might instead focus on increasing escapements at late chum salmon runs where eagles congregate in Southeast Alaska, British Columbia, and Northern Washington. Similarly, resource extraction that threatens late chum salmon runs would likely have a disproportionately large impact on regional eagle populations.

Supplemental Information

Supplemental Information 1 Raw camera trapping data from salmon spawning grounds

Click here for additional data file.

Supplemental Information 2 Summarized camera trap data from salmon spawning grounds

Click here for additional data file.

Supplemental Information 3 Camera trap data from roads and trails

Click here for additional data file.

We thank Rich Chapell from Alaska Department of Fish and Game for lending us a canoe and talking about salmon, Mike Howard for lending us a raft, and Marvin Willard and the village of Klukwan for providing logistical support. Thanks also to Jedediah Blum-Evitts for providing field assistance in 2013.

Additional Information and Declarations

Competing Interests

Author Contributions

The authors declare there are no competing interests.

Taal Levi and Rachel E. Wheat conceived and designed the experiments, performed the experiments, analyzed the data, wrote the paper, prepared figures and/or tables, reviewed drafts of the paper.

Jennifer M. Allen performed the experiments, reviewed drafts of the paper.

Christopher C. Wilmers conceived and designed the experiments, contributed reagents/materials/analysis tools, reviewed drafts of the paper.

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
