# Peer review of "Differential use of salmon by vertebrate consumers: implications for conservation"

_PeerJ, doi:10.7717/peerj.1157_

## Round 0.1 · original submission · Major Revisions

Your paper got 3 detailed reviews from people that actively work on bear-salmon interactions using camera traps. A wide variety of concerns are listed in their reviews.

Reviewer 1 suggests the experimental design is fatally flawed, whereas reviewer 2 simply says the design is unclear. I agree with the latter opinion. I also agree with the very positive reviewer 3 who notes that your paper contains a lot of interesting information that is valuable owing to the fact that behavior of bears and other consumers of salmon are poorly known.

So, I require a major revision in which you deal with all of the reviewers' comments. You don't have to agree with every point but you have to say how you dealt with each comment in your rebuttal letter. That will be a rather big job but I think the paper is worth it.

Reviewer 1 ·

Basic reporting

There are a few parts of the manuscript which are unclear, or are missing some detail. Here are a few examples that I noted, with some suggestions:

32- A quibble- Maybe replace marine nutrients with “marine energy and nutrients.” The nutrients are important for vegetation, but most vertebrates are more limited by energy. This is why bears will excrete most of the nitrogen in their urine.

36- Another citation for the diversity of species used by salmon- Shardlow and Hyatt, 2013, Ecological Indicators

38 – …for mobile consumers that move among runs… is maybe more clear

39-41- unclear

42- Change dispersion to distribution.

58- Although not conclusive, Schindler et al. 2013 suggests that bears also migrate to track resource pulses

76-78- You should note that the tendency for bears to “high grade” and only eat 25% of carcasses only occurs when salmon are both abundant and easy to catch. If not, bears will tend to eat almost everything. This is an important distinction because there is likely a threshold of salmon abundance below which bears will not leave anything for scavengers.

128- weighed-> weighted

225- “integrate over..” is a bit confusing. Maybe something like “bears maximize their annual energy balance by depositing fat during salmon abundance and hibernating during food scarcity”

Fig. 2- Missing the letter G, so the labels don’t match the fig. legend

Other- the spatiotemporal pattern of salmon spawning is important to understanding this paper. It might be helpful to show this with a multi-panel map showing where salmon are available during multiple time periods (e.g., summer, fall, winter). This would highlight the relative scarcity of salmon in late fall/winter.

Experimental design

Although I appreciate the objective of this paper, I don't think that the methods employed allow you to make the conclusions you make. The most important comments that address these issues are below (151, 171).

125-132- You are still overestimating the number of individuals using these areas, unless you are assuming that bears (and the other species) don’t hang around in one spot for more than 2 minutes. This is unreasonable given past research on the length of bear fishing bouts (Gende and Quinn, 2004; mean fishing bout length of 46 min. for dominant bears). If a bear fished for this duration in a location where it will trigger your camera, you might conclude that 23 unique bears visited the sites. However, I don’t think this is a problem if you re-frame what you are measuring. Rather than call them “unique visits” instead treat your data as an index of use of the salmon resource. Thus, a bear that walks past your camera will trigger it once, but is unlikely to stick around if there are no salmon. I think it is safe to assume that the number of detections correlates with the amount of use, even if it is the same few individuals.

133- Were carcasses placed at each camera throughout the entire season, or just for the period corresponding to the nearby run?

142-144- Here, with only a single carcass, the five minute threshold may discriminate between unique groups of bears, but a raven or magpie might take a long time to consume a carcass.

151- These results and interpretation hinge on three important assumptions that are not mentioned in the methods. 1) each species is equally likely to trigger the infrared motion sensor on the cameras; 2) camera trigger rates (detection rates) for a given species are equal across sites; and 3) the viewsheds of each camera are equal. Unfortunately, I think all of these assumptions are violated by the methods used. Assumptions 1 and 2 - If species-specific trigger rates were the same across all sites (a raven was equally likely to trigger a camera at a stream than on a river flat) then the violation of this assumption would not prevent comparing the pattern of use of ravens across habitats. However, when detection varies by habitat differently for different animals, this is a problem. At close range I expect all species to have similar change of triggering the cameras. For this reason I think the stream data is likely the best for comparing among species. At longer distances, however, an eagle may be more likely to trigger the camera than a magpie, making comparisons between species at these sites less reliable. Assumption 3- While this assumption is violated, it does not change your results as long as you stick to comparing among species at a single site, as you do for ravens on line 159.

149- this variation could be because of variation in motion triggers because of different distances to cameras rather than variation in patterns of use.

171- In this paragraph you implicitly use the detection rates of your road-side cameras as estimates of relative abundance of several species within your study area. I don’t think this is valid; relative detection rates may vary as a function of differences in habitat selection among species. For example, black and brown bears may differentially avoid roads due to the risk of running into people. This would be better supported if you had a rough estimate of black vs. brown bear abundances for your study area.

Validity of the findings

Here are some specific notes.

42-43- Yes, integration of wildlife and salmon management is very important.

153-154 This is likely a function of availability. Even though bears are less efficient when fishing at lake shores, they will if those are the only salmon available. Was this the case? If not, subordinate bears or sows with cubs may choose a poorer fishing habitat to avoid intraspecific competition or infanticide (Ben-David et al. 2004).

163-164- Okay because you are making a relative comparison and you were detecting ravens throughout the period, and I’d expect ravens, gulls, magpies and stellar’s jays to all have the same likelihood of triggering the cameras.

167-170- I don’t think the data supports this interpretation. Yes, there were more brown bears detected at Herman creek, but Herman creek had similar numbers of eagles, more ravens, and more magpies than Klehini flats. I think it would be more accurate to say that there were more bears detected at Herman creek than Klehini flats, however, you still have the problem of differences in the size of camera viewsheds. This would be more clear if the x- axes of figure 3 were all the same. Again, comparisons across species and sites might be due to a confounding relationship between species-specific detection probability and site.

173-177- Yes, and this result agrees with some past research on the subject (Fortin et al. 2007).

197-202- Nice, this addresses my earlier comment.

203- I like that you address the implications of a multi-salmon species system on consumers, but I would be careful about saying that you can fish the less valued species more intensively because they are all equally used by bears. Gende et al. (2004, Oikos) documented bears selecting salmon with higher fat content because they were a more beneficial resource; my personal observation is that bears prefer kings> coho> sockeye> chums> pinks. Assuming this reflects the nutritional benefit of each species, this suggests that you cannot exchange a coho for a pink and not harm bears. Another interpretation of your results is that bears and human fishermen likely benefit from the diversity of salmon species in this area. Schindler et al. (2010) showed how like life history diversity within sockeye salmon populations created a population portfolio that benefited predators and commercial fishers. Run timing diversity extended the duration of salmon availability to wildlife consumers and annual returns were more stable for wildlife and commercial fisheries because of the so called “portfolio effect.” These effects may be even more beneficial to consumers in your system because it involves multiple salmon species.

209- 210- Yes, a perfect example of how multiple salmon species system can extend the duration of access to salmon for bears.

213- This paragraph is interesting.

224- Or, are the other species just less conspicuous?

233-235- nice

238-243- Salmon are likely important to mink population productivity (Ben David, 2011), but their use of salmon is more difficult to observe than a bear or eagle.

248-251- I strongly disagree. Salmon species are not equally valuable as a resource, nor are salmon spawning in different areas equally available. You are assuming that the number of detections of an bear at a location is always correlated with the value of the resource. Contrast the number of times you would detect a bear that spends an hour to catch a sockeye salmon spawning on a lake shore versus a bear that can easily catch pink salmon in a shallow stream. The bear at the stream may become satiated in 45 minutes and go take a nap while the other bear fishes for hours. You would detect the lake shore bear many more times, but the pink salmon population is clearly more valuable to the bear.

253-256- Good point.

Additional comments

Because of problems with the experimental design, I don't think that the findings are valid. I think this paper could be rewritten to focus just on contrasting the patterns of detection of bears and eagles across time (the other species distract from the clear contrast between bears and eagles). The problems with the methodology (comment 151 above) can be minimized if you pooled detections across sites with a given period of availability (e.g. late chum, early pink) and normalized the number of detections between zero and one (divide by max daily detections of a species) to account for differences in probability of detection. With these changes I think you could contrast the patterns of use (bears used salmon during all time periods, while eagles focused on late chum, reflecting their different life history strategies).

If this study is ongoing, it would be better to use time lapse photography instead of motion trigger, or, pair a motion trigger camera with a video camera to check whether detection probability varies by species and/or site.

Reviewer 2 ·

Basic reporting

Valuable research question.

Abstract well-written.

Main text needs a bit more proof-reading:
ex., line 146 "weighed" should be "weighted".
ex., line 25-26 change "to determine how different wildlife species use salmon resources" to something along the lines of "to discern potentially different use patterns among consumers"

Line 87: It is not exactly the brains (rather small) bears are selectively eating but more accurately the fat deposits on the top portion of their heads.

Lines 89-90: consuming on 25% of caught salmon is relatively rare. When salmon are very abundant and bears are relatively satiated they do often tend to "high-grade" but quite often come back to carcasses they have high-graded and left. This is more common after about two weeks into the salmon run. Early on "high-grading" is much less common.

Experimental design

Need to be better state what the limitations and assumptions are with methodology & design.

Design and methodology needs more thought and possible improvement.

Why was motion-detection used instead of a more systematic time interval sampling regime? Different consumer species are likely to present different rates of triggering of cameras. This probably biased detection rates and results more than a time lapse photographic approach. What was the layout of the twenty cameras used to monitor wildlife activity on spawning grounds? How were cameras distributed among and within different spawning grounds categories?

How was run timing and salmon abundance measured? General observations or a more systematic approach. Needs to be clearly stated.

The assumption that unique individuals were detected by two or five minute delays is probably not true. .....

Run tests for significance of differences among among consumer types and among different sites or spawning categories.....

Stated that placed cameras along roads and trails in early summer to provide relative index of abundance of various terrestrial carnivores: Limitations of this is that use of trails and roads may not be proportional across species (some species tend to use roads and established trails more than others); only sampling or indexing use in early summer may have biased results because distribution of different terrestrial carnivores varies according to season and trails and roads are not randomly distributed across the study area….. Might want to also see if ADFG or Forest Service may have accurate estimates of terrestrial carnivore abundance. .....

Lines 161-163: The assumption that a five minute delay will confidently unique visitations (individuals?) is likely not true - at least for many consumers.

Validity of the findings

The authors have a wealth of photographic data that is valuable. I suggest they try using more refined methods and analyses to make the most of this information.

Lines 174-177: Occurrence or frequency of visitation at various sites does not equate to "higher foraging efficiency".

Reviewer 3 ·

Basic reporting

.

Experimental design

.

Validity of the findings

.

Additional comments

General Comments

This paper makes natural history observations about the use of salmon resources by terrestrial consumers. The observations, though relatively basic, are valuable to our understanding of the direct effects of salmon subsidies. I look forward to seeing this work published and I hope to see more basic natural history work given some of the holes in our understanding of salmon subsides.

Since this paper is putatively about vertebrate consumers, it would be helpful if it would make a small effort to consider the work that has been done on vertebrate consumers that lack feathers and fur. Stream ecologists have done most of the work on salmon subsidies, and they may be confused by the terminology in this manuscript and its tendency to ignore prior work in freshwater environments. For example, the list of vertebrate consumers in the intro does not include fishes! I realize that PeerJ is about getting the data out there and less about placing results in a broader context, but as long as the format is going to include a results and discussion section, I think it’s reasonable to expect manuscripts to make a decent attempt at placing new material within the context of existing work. See comments below for specific examples.

While I agree that this paper has some management implications, the discussion does not put forward arguments that are either logically strong, or well supported by existing work. The paper makes little attempt to reference existing work that considers how fisheries can affect salmon availability across space and time (i.e. Doctor et al. 2010 Trans. Am. Fish Soc., Boatright et al. 2004 Trans. Am. Fish. Soc.) or how salmon abundance affects consumer foraging opportunities (Bentley et al. 2012 Ecosphere).

Specific Comments

12: “distinct life histories” seems like an overstatement, many salmon consumers opportunistic foragers
34: what about fishes?
39: I didn’t totally follow this sentence
48: Smaller birds have higher caloric demands per unit body mass
51: the phrase distinct life-histories is used a few times but I don’t get much meaning out of it
58: Don’t eagles also deposit fat (though not to the same degree) and don’t bears also track asynchronous run-timings? Lisi et al. 2012 Geomorphology showed that substantial phenological variation in salmon can be expressed over small spatial extents, within the range of consumers with modest mobility. In fact due to limited spatial autocorrelation in salmon phenology, the relationship between spatial extent and salmon phenological extent may asymptote within a bear’s home range.
64: I’ve seen >70 bears on a 3 km stream with the last run of salmon in a watershed, I’m not sure that late season aggregations are unique to eagles.
69: the effects of stream size on bear predation is well documented by Quinn, Carlson, Hendre and others and could be cited here.
69: Habitat features have been shown to strongly influence whether fishes can exploit salmon subsidies: Armstrong et al. 2010 Ecology Thermal heterogeneity mediates the effects of pulsed subsidies across a landscape…
83: What is a pool off of a larger river? Floodplain habitat?
89: a sentence reiterating the methodologies would be helpful, so that we know what monitoring an individual carcass means.
93: I’d never heard of lake-spawning coho salmon, are you sure they are spawning along the lake sure and not simply aggregating there? Interesting regardless.
96: How were the salmon breeding phenologies determined?
114: Not a big deal, but usually freshwater scientists don’t call small bodies of flowing water “creeks”, they call them “streams” and only use creek in the name.
116: It’d be nice if you could define what you mean by a pool in terms that would make sense to a stream ecologist – the current definition is likely not what stream ecologists think of as a pool. The results of this paper are relevant to freshwater folks studying salmon subsidies, so to maximize the impact it would help to use clear terminology.
139: What was the condition of the carcass or how were they aquired? Since the energy density of spawning salmon declines over time (Hendry and Berg 1999 Can. J. Zool) the state of the carcass could influence its appeal to scavengers.
146: I’m not sure I find the total aggregate # of images useful, but since it’s listed I’m curious of whether false triggers are included and whether the cameras worked continuously and were never taken out by bears or ran out of batteries.
154: Quinn et al. 2001 do not suggest that bears do not feed along large rivers or lake shores, but instead that they exert lower predation rates on salmon in these habitats. I don’t think that the observation that bears still show up at rivers and lakes shores will be surprising to anyone, and it has also been documented in the literature. I’m not saying that this documentation is not a worthy contribution for natural history, just that it shouldn’t be pitched as contrary to the results of Quinn et al. 2001 at least not without more nuance.
170: The tricky part about making inferences from these data is that we don’t know all the salmon foraging habitats that were within the foraging neighborhoods of these consumers—if the birds didn’t show up at a small stream with a camera on it, was that because they don’t like small streams, or because they were on another small stream where a camera was not present? I’m not saying that these data don’t provide insights, but if the authors could address this potential issue it would make their arguments stronger.
185: If brown bears are quickly grabbing many of the carcasses, how does that affect the inference of scavenging by other consumers? Would you see potentially different results if you surrounded the carcasses with electric fences permeable to small carnivores?
192: What would prevent eagles from foraging in forested habitats? I’m just curious since they seem invulnerable to predation from all but bears, but they do seem to prefer open habitats.
197: I’ve noticed that the time of year strongly influences whether bears will consume carcasses left as bait. In unpublished studies of maggot scavenging dynamics, bears would grab a carcass within a couple hours at the very beginning of the salmon runs. In contrast (and intuitively) during the peak of the runs, most carcasses left out were not scavenged.
203: I am skeptical that managers would get much traction by arguing that a recent study showed bears fed on a variety of salmon runs, and therefore fishing levels should decline. The argument for backing off of chum and pink runs is interesting and perhaps ecologically valid, but those aren’t necessarily “low value” species. They may be on a per pound basis, but certainly the aggregate value of pink and chum fisheries can be substantial. Further, many Alaskans (whether native, Caucasian, “urban”, and rural) have little interest in maximizing brown bear productivity, but are very concerned about salmon yields.
213: Could the absence of consumers on early runs be related to other factors, such as constraints due to reproduction, or use of alternative food sources? For example some herbivores catch the “green wave” late and leave it early, presumably due to trade-offs between resource tracking and arrival at breeding sites (Kolzsch et al. 2015 J Anim. Ecol.)
228: it’s not clear whether this is speculation or backed by existing data or studies
234: the term integrate is confusing when applied to energy allocation
242: Seems worth noting that salmon carcasses are likely much lower in energy density than the alternative prey of small carnivores—a spawned out salmon is about ~3 kJ/g (Hendry and Berg 1999) whereas a rodent is about 2-3-times as energy dense (Cox and Secor 2007, Comp. Biochem. Phys A). Foraging theory suggests that the abundance of preferred prey (not alternative prey) influences prey switching, which in turn suggests that small carnivores might only switch to salmon when rodent cycles are in their low phase. See Lisi et al. 2013 Ecology of Freshwater Fish for twist on fish switching to rodent prey during years with low salmon abundance.
248: Combining all salmon biomass when managing for salmon misses the point that certain salmon populations have disproportionate importance to consumers, which I thought was a key point of this paper. I’m not saying it’s feasible to manage salmon fisheries for specific populations, but it would certainly seem important to avoid management practices that are likely to hammer specific populations, for example the common practice of meeting escapement goals and then heavily harvesting late arriving fish.
255: these results seem overstated, I would say “would likely” but “will” is not demonstrated by these data.

Figure 1: Can you color or symbol the sites by run timing and or species

---

## Round 0.2 · accepted · Accept

The reviewers were satisfied with your revisions, as was I. Please do consider the comments of reviewer 4 which will improve the paper.

Reviewer 1 ·

Basic reporting

No additional comments

Experimental design

No additional comments

Validity of the findings

No additional comments

Additional comments

The authors have addressed all of my concerns either through changes to the manuscript or clarification in their rebuttal. I recommend that the article is accepted for publication.

Reviewer 4 ·

Basic reporting

No Comments

Experimental design

No Comments

Validity of the findings

Line 275: I still have a few issues with the implication that chum and pink fisheries are less important to commercial fishers because of their lower price. Yes, pinks and chums are not worth much per pound, but they are the most abundant species of salmon—several times more abundant than Chinook. It’s the price per pound * harvestable biomass that determines the value of the fishery. Further these stocks could be of particular significance to humans for the same reason they are to wildlife, for example if they provide commercial fishing opportunities at unique times of the year, or allow fishers to mitigate for low returns of other salmon stocks.

Additional comments

Comments on Intro:
56: The body size argument isn’t very compelling to me, but that’s OK.
69: I still find this paragraph inaccurate. The paragraph implies that bears and eagles have fundamentally different tactics for exploiting salmon resource pulses. I would argue that they exhibit very similar behavior, just expressed at different spatial and temporal scales due to the constraints of their life-history and physiology. Bears track asynchronous salmon runs to build their fat stores. Suggesting they don’t need to track asynchronous resource pulses because they can store fat is inaccurate. This paragraph would be much clearer if it stated up front that several species exhibit behavioral tactics to exploit resources pulses, tracking asynchronous salmon runs to protract their access to this locally ephemeral food source (Ruff et al. 2010 Ecology, Schindler et al. 2013, additional eagle paper if there is one) then say that the spatial and temporal extent of this behavior depends on the species life-history/physiology/etc.

I don't understand why the importance of the salmon resource wave to bears is downplayed in the intro and emphasized in the discussion.

Reviewer 5 ·

Basic reporting

No additional comments.

Experimental design

No Comments.

Validity of the findings

No Comments.

Additional comments

I am satisfied with the changes and explanations provided. I think this article will make a valuable contribution. Very interesting. Recommend highly for publication.

---

## Author Rebuttal · Round 0.2

Dr. Taal Levi, Assistant Professor
Department of Fisheries and Wildlife
Oregon State University, 104 Nash Hall, Corvallis, Oregon 97331-3803
T 541.737.4067 | F 541.737.3590 | E Taal.Levi@oregonstate.edu
fw.oregonstate.edu

5 June 2015

Dear Dr. Stanford,

Thank you very much for your evaluation of our manuscript entitled "*Differential use of salmon by vertebrate consumers: implications for conservation.*"  We appreciate the thorough and detailed reviewer comments, which have improved the manuscript. This manuscript is very much focused on natural history aspects of terrestrial salmon consumers. We have further limited our inferences and discussion and feel that the experimental design is adequate to justify our results and discussion. There are certainly biases involved with camera trapping, but the broad-scale patterns that we observe are interesting and generate hypotheses about which terrestrial vertebrates benefit most from pulses of salmon availability as a function of spatiotemporal availability.

We would like to resubmit the revised manuscript as a research article for publication in PeerJ. Responses to specific reviewer comments are below. Please let us know if you have any additional concerns, questions, or edits.

Sincerely,

Taal Levi

# Reviewer Comments

## Reviewer 1 (Anonymous)

**Basic reporting**

There are a few parts of the manuscript which are unclear, or are missing some detail. Here are a few examples that I noted, with some suggestions:

32- A quibble- Maybe replace marine nutrients with "marine energy and nutrients." The nutrients are important for vegetation, but most vertebrates are more limited by energy. This is why bears will excrete most of the nitrogen in their urine.

*Indeed. Done.*

36- Another citation for the diversity of species used by salmon- Shardlow and Hyatt, 2013, Ecological Indicators

*We had cited Shardlow & Hyatt elsewhere, but we now cite here as well*

38 – …for mobile consumers that move among runs… is maybe more clear

*Done.*

39-41- unclear

*This has been changed for clarity.*

42- Change dispersion to distribution.

*Done.*

58- Although not conclusive, Schindler et al. 2013 suggests that bears also migrate to track resource pulses

*Bears certainly move between feeding patches to take advantage of asynchronous salmon availability, but this behavior is distinct from the regional movements of eagles (we have 12 GPS tagged eagles on air) that move between Southern BC, all of SE Alaska, and at times into the Northern Yukon and interior Alaska. When food is no longer locally abundant, bears hibernate whereas eagles move. Similarly, the two collared bears on the Chilkoot River do not move to the Chilkat to feed despite the presence of a very late chum and coho run on the Chilkat (which doesn't freeze due to groundwater upwelling in the river). We suggest that the degree to which this occurs is likely to be context dependent and requires further research before results from Bristol Bay can be extrapolated to the rugged terrain of Southeast Alaska.*

76-78- You should note that the tendency for bears to "high grade" and only eat 25% of carcasses only occurs when salmon are both abundant and easy to catch. If not, bears will tend to eat almost everything. This is an important distinction because there is likely a threshold of salmon abundance below which bears will not leave anything for scavengers.

*Absolutely. Done.*

128- weighed-> weighted

*Thank you. Fixed.*

225- "integrate over.." is a bit confusing. Maybe something like "bears maximize their annual energy balance by depositing fat during salmon abundance and hibernating during food scarcity"

*Good suggestion! We hope you don't mind that we used your suggestion verbatim.*

Fig. 2- Missing the letter G, so the labels don't match the fig. legend

*Thank you. Fixed*

Other- the spatiotemporal pattern of salmon spawning is important to understanding this paper. It might be helpful to show this with a multi-panel map showing where salmon are available during multiple time periods (e.g., summer, fall, winter). This would highlight the relative scarcity of salmon in late fall/winter.

*As suggested by Reviewer 3, we have used colors to indicate the species of salmon, and we now use the figure legend to describe the timing of salmon availability.*

**Experimental design**
Although I appreciate the objective of this paper, I don't think that the methods employed allow you to make the conclusions you make. The most important comments that address these issues are below (151, 171).

125-132- You are still overestimating the number of individuals using these areas, unless you are assuming that bears (and the other species) don't hang around in one spot for more than 2 minutes. This is unreasonable given past research on the length of bear fishing bouts (Gende and Quinn, 2004; mean fishing bout length of 46 min. for dominant bears). If a bear fished for this duration in a location where it will trigger your camera, you might conclude that 23 unique bears visited the sites. However, I don't think this is a problem if you re-frame what you are measuring. Rather than call them "unique visits" instead treat your data as an index of use of the salmon resource. Thus, a bear that walks past your camera will trigger it once, but is unlikely to stick around if there are no salmon. I think it is safe to assume that the number of detections correlates with the amount of use, even if it is the same few individuals.

*We apologize that this was confusing. We were indeed trying to create an index of abundance. Many papers report pictures per camera-day, or some other photo capture rate. We are reporting a similar index, but we are consolidating dozens or hundreds of photos from one foraging bout into a single "unique" encounter. The word unique was never meant to indicate a unique individual, but rather a unique encounter or camera trigger. We have now clarified this to read:*

"We used camera trapping rates as an index of abundance. To avoid overestimating visitation rates when the same individual or group foraged in front of the camera for an extended period, we post-processed the camera data to identify unique visitations. We defined unique group visitations (where "unique" refers to a unique encounter rather than a unique individual) as visits with a greater than two minute delay between the last photograph from one visit to the first photograph of the next."

133- Were carcasses placed at each camera throughout the entire season, or just for the period corresponding to the nearby run?

*We have now clarified that the carcasses were locally collected and used during the period of the run. The intent was to see which species benefit from salmon carcasses deposited on land, as occurs when bears deposit partially consumed carcasses,which required collection of carcasses from adjacent spawning areas.*

142-144- Here, with only a single carcass, the five minute threshold may discriminate between unique groups of bears, but a raven or magpie might take a long time to consume a carcass.

*As above, we are only treating "unique" visitations as an index of abundance with no assertion that this represents unique individuals. If a raven spends ten minutes in front of the camera, leaves for more than five minutes, and then returns, we would count this as two unique raven encounters.*

151- These results and interpretation hinge on three important assumptions that are not mentioned in the methods. 1) each species is equally likely to trigger the infrared motion sensor on the cameras; 2) camera trigger rates (detection rates) for a given species are equal across sites; and 3) the viewsheds of each camera are equal. Unfortunately, I think all of these assumptions are violated by the methods used. Assumptions 1 and 2 - If species-specific trigger rates were the same across all sites (a raven was equally likely to trigger a camera at a stream than on a river flat) then the violation of this assumption would not prevent comparing the pattern of use of ravens across habitats. However, when detection varies by habitat differently for different animals, this is a problem. At close range I expect all species to have similar change of triggering the cameras. For this reason I think the stream data is likely the best for comparing among species. At longer distances, however, an eagle may be more likely to trigger the camera than a magpie, making comparisons between species at these sites less reliable. Assumption 3- While this assumption is violated, it does not change your results as long as you stick to comparing among species at a single site, as you do for ravens on line 159.

*These points apply to the monitoring of salmon spawning grounds, but not to the monitoring of salmon carcasses, which were always monitored at close range. We monitored spawning grounds for one year and salmon carcasses for two years. For spawning grounds, violations of these assumptions clearly add noise, but the purpose of this paper is not to precisely quantify. This is a natural history paper describing the species that visit different types of salmon spawning areas. We recognize the limitations of camera trapping, which is why we make no pretense to use statistical tests on the camera trapping rates but simply provide descriptive results. Many of the results were not intuitive when we began monitoring and are not well represented in the literature. For example, when we began working in this system, we did not expect the level of bear-dominance that we observed on most spawning areas because the literature stresses that such a large diversity of consumers use salmon. Similarly, we did not expect such patchy use of salmon by eagles and gulls, which completely avoided feeding on pink salmon in forested streams. Ravens, on the other hand, did consume pink salmon in these streams, which is intuitive because they are much more of a forest bird. We additionally expected coyotes to be more active salmon consumers. Finally, we expected vertebrates to use carcasses in the forest at higher rates, and we did not expect that visitation to these forest carcasses would also be so bear-dominated.*

*If this paper, for example, tried to use camera trapping results to quantitatively infer how much marine derived energy is flowing into each species of primary consumer, then the reviewer's points would be spot on because the quantitative biases would cause fundamental errors. However, our qualitative results are robust to the assumptions pointed out by the reviewer, and they conform with our intuition and many hours spent observing these salmon spawning grounds with both eyes and cameras. While our results are not precisely quantitative, these natural history observations would be very useful to, for instance, young naturalists/scientists interested in terrestrial vertebrate-salmon linkages. We feel there is a need for more natural history and description in ecology and that PeerJ is an excellent venue for such papers.*

*We now more explicitly state the limitations of motion-activated cameras in the Methods section:*
"Motion-activated cameras on spawning areas may produce biased results because larger-bodied animals can trigger the camera at a greater distance in the viewshed. We elected not to sample on a systematic interval because such sampling is likely to miss infrequent visitors unless the sampling frequency is very high (i.e. nearly continuous monitoring), which would produce an intractably large number of photos for a study of this scale. Additionally, we were not concerned about any bias introduced by motion-activated cameras because this research was designed to understand qualitative and broad-scale natural history patterns."

149- this variation could be because of variation in motion triggers because of different distances to cameras rather than variation in patterns of use.

*See above.*

171- In this paragraph you implicitly use the detection rates of your road-side cameras as estimates of relative abundance of several species within your study area. I don't think this is valid; relative detection rates may vary as a function of differences in habitat selection among species. For example, black and brown bears may differentially avoid roads due to the risk of running into people. This would be better supported if you had a rough estimate of black vs. brown bear abundances for your study area.

*Certainly it is very telling that black bears were so abundant on roads and trails but nearly absent on salmon spawning grounds, which is the result we are conveying using encounter rates as an index of abundance. There are no black bear density estimates from our study site, and in fact very few anywhere from mainland Southeast Alaska. One exception is Kyle Pinjuv's 2013 MS thesis, which estimated black bear densities near Gustavus. Brown bears are just recently recolonizing and black bears are relatively abundant at an estimated density of 27.3 per 100km$^2$. However, recent research by the Alaska Department of Fish and Game from watersheds emptying into Berners Bay, which is a nearby system with similar glacial morphology, found brown bear densities of 45.3 per 100km$^2$ and black bear densities of 58.3 per 100km$^2$ despite a research protocol designed to target brown bears (Kevin White Alaska Department of Fish and Game, personal communication). Additionally, black bears are more heavily hunted at our field site as a subsistence resource, including over bait. In contrast, brown bears are not hunted for food but rather for trophies and not over bait. Trophy hunting of brown bears typically occurs at salmon spawning grounds. It is not inconceivable that brown bears avoid roads and trails more than black bears, but the risk posed by hunting would suggest that, if anything, the opposite is more likely. However, given the light use of dirt roads and trails in this remote location, and the extensive use of roads and trails by bears that is evident from scat surveys and camera trapping, we do not think that there is any significant bias due to avoidance of humans. Other common monitoring protocols have similar biases, and we are*

*confident in the results that demonstrate that (a) black bears are very abundant, and (b) no black bears were observed eating salmon.*

**Validity of the findings**
Here are some specific notes.

42-43- Yes, integration of wildlife and salmon management is very important.

153-154 This is likely a function of availability. Even though bears are less efficient when fishing at lake shores, they will if those are the only salmon available. Was this the case? If not, subordinate bears or sows with cubs may choose a poorer fishing habitat to avoid intraspecific competition or infanticide (Ben-David et al. 2004).

*We have removed this text in response to comments from another reviewer.*

163-164- Okay because you are making a relative comparison and you were detecting ravens throughout the period, and I'd expect ravens, gulls, magpies and stellar's jays to all have the same likelihood of triggering the cameras.

167-170- I don't think the data supports this interpretation. Yes, there were more brown bears detected at Herman creek, but Herman creek had similar numbers of eagles, more ravens, and more magpies than Klehini flats. I think it would be more accurate to say that there were more bears detected at Herman creek than Klehini flats, however, you still have the problem of differences in the size of camera viewsheds. This would be more clear if the x- axes of figure 3 were all the same. Again, comparisons across species and sites might be due to a confounding relationship between species-specific detection probability and site.

*We should have made clear that we were comparing Herman Creek to both Chilkat and Klehini Flats. Chum salmon spawn early at Klehini Flats with a much smaller run than at Chilkat Flats and Herman Creek. Nevertheless, the reviewer is correct that this is too speculative. Eagles and magpies certainly fed extensively at Herman Creek. What was very clear is that gulls barely fed at this forested site or any other. We have changes this text to read:*

"The absence of birds, other than ravens, at early salmon runs was followed by a high concentration of eagle and gull activity at late salmon runs (Figs. 3-4). However, avian scavengers fed on salmon less often than brown bears at the late chum salmon run at Herman Creek. We speculate that stream morphology and run timing both influenced where these consumers preferentially fed on salmon (Figs 3-4). This was particularly evident by the very few gulls observed at Herman Creek relative to Klehini Flats and Chilkat Flats."

*We initially made the figures with equal x-axis scales across panels, but this obscured the within site relationships. The current scale made the results easier to visualize and interpret.*

173-177- Yes, and this result agrees with some past research on the subject (Fortin et al. 2007).

*Indeed. We now cite Fortin et al. 2007 here.*

197-202- Nice, this addresses my earlier comment.

203- I like that you address the implications of a multi-salmon species system on consumers, but I would be careful about saying that you can fish the less valued species more intensively because they are all

equally used by bears. Gende et al. (2004, Oikos) documented bears selecting salmon with higher fat content because they were a more beneficial resource; my personal observation is that bears prefer kings> coho> sockeye> chums> pinks. Assuming this reflects the nutritional benefit of each species, this suggests that you cannot exchange a coho for a pink and not harm bears. Another interpretation of your results is that bears and human fishermen likely benefit from the diversity of salmon species in this area. Schindler et al. (2010) showed how like life history diversity within sockeye salmon populations created a population portfolio that benefited predators and commercial fishers. Run timing diversity extended the duration of salmon availability to wildlife consumers and annual returns were more stable for wildlife and commercial fisheries because of the so called "portfolio effect." These effects may be even more beneficial to consumers in your system because it involves multiple salmon species.

*We have removed some of this speculation in response to another reviewer comment. On a per-capita basis, your observation about the hierarchy of bear preference may be true, but this doesn't seem to play out on a landscape scale where pink/chum systems can support some of the highest densities of bears (e.g. Admiralty). We speculate that this is because pink and chum tend to be more accessible to bears than king and coho so that there is a tradeoff between accessibility and nutritional content.*

*The point that we are trying to make is that pink, and to a lesser degree chum, are worth so little to fishermen on a per-capita basis, but are quite important to bears. These species also have early and late run timing, which extends the period of salmon availability. A strategy to maximize the benefit to bears while minimizing economic impact might include managing for higher pink and chum escapement. We have changed this text to read:*

"The extensive use of all salmon resources by brown bears suggests that bears are benefiting from a diversity of run timing by moving between asynchronous spawning aggregations to maximize their nutritional intake (Schindler et al. 2013). It is possible that relaxing harvests on lower value salmon species such as pink salmon (2014 Southeast Alaska ex-vessel value of $0.28/lb) and chum ($0.64/lb), while fishing higher value species such as king ($4.27/lb), and sockeye ($1.75/lb) for maximum sustainable yield would be an effective strategy to increase bear population productivity with less economic impact. This seems particularly plausible because pink and chum have early and late run times respectively, which extends the temporal availability of salmon biomass, although they do not permeate as far into watersheds as other salmon species, restricting access to more interior salmon predators and scavengers."

*Increasing escapement of other salmon species would certainly benefit bears as well, but there is a disproportionate economic impact to reducing harvests. For instance, the 37 million pink salmon harvested in Southeast Alaska in 2014 had approximately the same total value as the 3.7 million coho salmon that were harvested.*

209- 210- Yes, a perfect example of how multiple salmon species system can extend the duration of access to salmon for bears.

*Indeed.*

213- This paragraph is interesting.

224- Or, are the other species just less conspicuous?

233-235- nice

238-243- Salmon are likely important to mink population productivity (Ben David, 2011), but their use of salmon is more difficult to observe than a bear or eagle.

*Certainly, but this statement is meant to convey that their nutritional requirements are lower because they are small bodied, and unlike similarly sized birds they do not congregate in large numbers. Smaller amounts of salmon that are widely dispersed on the landscape allow more individual animals to consume salmon, as opposed to large spawning aggregations that are in surplus to the resident individual mink.*

248-251- I strongly disagree. Salmon species are not equally valuable as a resource, nor are salmon spawning in different areas equally available. You are assuming that the number of detections of an bear at a location is always correlated with the value of the resource. Contrast the number of times you would detect a bear that spends an hour to catch a sockeye salmon spawning on a lake shore versus a bear that can easily catch pink salmon in a shallow stream. The bear at the stream may become satiated in 45 minutes and go take a nap while the other bear fishes for hours. You would detect the lake shore bear many more times, but the pink salmon population is clearly more valuable to the bear.

*We agree that this is too strong a statement. We have changed this to:*
"Understanding how salmon use is influenced by consumer life history, run timing and stream morphology, is an important first step toward integrating wildlife needs into salmon management as part of an ongoing paradigmatic shift toward Ecosystem-based fisheries management (Levin et al. 2009). Although there are quantitative differences in the nutritional quality of salmon and their accessibility, all salmon on the landscape were heavily used by bears. In contrast, the activity pattern of bald eagles suggests that fisheries management that considers their nutritional requirements might instead focus on increasing escapements at late chum salmon runs where eagles congregate in Southeast Alaska, British Columbia, and Northern Washington."

253-256- Good point.
**Comments for the author**
Because of problems with the experimental design, I don't think that the findings are valid. I think this paper could be rewritten to focus just on contrasting the patterns of detection of bears and eagles across time (the other species distract from the clear contrast between bears and eagles). The problems with the methodology (comment 151 above) can be minimized if you pooled detections across sites with a given period of availability (e.g. late chum, early pink) and normalized the number of detections between zero and one (divide by max daily detections of a species) to account for differences in probability of detection. With these changes I think you could contrast the patterns of use (bears us ed salmon during all time periods, while eagles focused on late chum, reflecting their different life history strategies).

If this study is ongoing, it would be better to use time lapse photography instead of motion trigger, or, pair a motion trigger camera with a video camera to check whether detection probability varies by species and/or site.
# Reviewer 2 (Anonymous)
**Basic reporting**
Valuable research question.

Abstract well-written.

Main text needs a bit more proof-reading:
ex., line 146 "weighed" should be "weighted".

*Fixed. Thank you.*

ex., line 25-26 change "to determine how different wildlife species use salmon resources" to something along the lines of "to discern potentially different use patterns among consumers"

*Good suggestion. Changed.*

Line 87: It is not exactly the brains (rather small) bears are selectively eating but more accurately the fat deposits on the top portion of their heads.

*In response to another reviewer comment, this has been changed to* "After capturing salmon in rivers and streams, brown bears move carcasses to land to feed, often feeding selectively on energy-rich body parts when salmon are abundant and easy to catch"

Lines 89-90: consuming on 25% of caught salmon is relatively rare. When salmon are very abundant and bears are relatively satiated they do often tend to "high-grade" but quite often come back to carcasses they have high-graded and left. This is more common after about two weeks into the salmon run. Early on "high-grading" is much less common.

*We observed this frequently and talk about this in the discussion section. We are not aware of papers discussing this behavior, but absolutely, bears were the most common species to scavenge the high-graded salmon carcasses!*

**Experimental design**
Need to be better state what the limitations and assumptions are with methodology & design.
Design and methodology needs more thought and possible improvement.

*Please see comments to reviewer 1.*

Why was motion-detection used instead of a more systematic time interval sampling regime? Different consumer species are likely to present different rates of triggering of cameras. This probably biased detection rates and results more than a time lapse photographic approach. What was the layout of the twenty cameras used to monitor wildlife activity on spawning grounds? How were cameras distributed among and within different spawning grounds categories?

*We now more explicitly state the limitations of motion-activated cameras in the Methods section:*
"Motion-activated cameras on spawning areas may produce biased results because larger-bodied animals can trigger the camera at a greater distance in the viewshed. We elected not to sample on a systematic interval because such sampling is likely to miss infrequent visitors unless the sampling frequency is very high (i.e. nearly continuous monitoring), which would produce an intractably large number of photos for a study of this scale. Additionally, we were not concerned about any bias introduced by motion-activated cameras because this research was designed to understand qualitative and broad-scale natural history patterns."

How was run timing and salmon abundance measured? General observations or a more systematic approach. Needs to be clearly stated.

*We now state that* "We identified all run timings through direct observation while visiting salmon spawning areas identified by Alaska Department of Fish and Game"

The assumption that unique individuals were detected by two or five minute delays is probably not true.

*See comments to reviewer 1. We are sorry for the confusion. We are using camera trap rates as an index of abundance, but we wanted to avoid counting all of the 100s of pictures taken in a single feeding bout. By adding a delay, our intention is to make this a better index of abundance. So "unique" refers to a unique "encounter" but not a unique bear.*

Run tests for significance of differences among among consumer types and among different sites or spawning categories.....

*We specifically did not run statistical tests because of the limitations of camera trapping, which were also identified by the reviewers. The indices of abundance are not comparable across sites, primarily because the different viewsheds among camera sets, so that cross-site comparisons of mean camera trapping rates are not possible. We now reiterate that this paper is a natural history paper that describes our observations.*

Stated that placed cameras along roads and trails in early summer to provide relative index of abundance of various terrestrial carnivores: Limitations of this is that use of trails and roads may not be proportional across species (some species tend to use roads and established trails more than others); only sampling or indexing use in early summer may have biased results because distribution of different terrestrial carnivores varies according to season and trails and roads are not randomly distributed across the study area….. Might want to also see if ADFG or Forest Service may have accurate estimates of terrestrial carnivore abundance. .....

*See response to reviewer 1. Unfortunately there are no terrestrial carnivore density estimates for this area, but there are brown and black bear density estimates from nearby systems (see above). We are in regular communication with Rod Flynn and Anthony Crupi from the Douglas Office of ADFG. A first bear project may soon be starting in the Haines area. Kevin White from the Douglas ADFG office has collared two wolves, the first ever from this Northern Southeast Alaska, which is ecologically distinct because moose are the only ungulate prey in contrast to the deer systems throughout most of the rest of this region.*

*We have added the following text to the methods to qualify the results:*
"This method may introduce biases if some carnivores select for or against roads and trails, but encounter rates serve as a useful index of abundance."

Lines 161-163: The assumption that a five minute delay will confidently unique visitations (individuals?) is likely not true - at least for many consumers.

*See response to reviewer 1. We are only using an index of abundance.*

**Validity of the findings**
The authors have a wealth of photographic data that is valuable. I suggest they try using more refined methods and analyses to make the most of this information.

*We feel that the inference that can be made from such data are restricted to natural history observations. See above*

Lines 174-177: Occurrence or frequency of visitation at various sites does not equate to "higher foraging efficiency".

*We have removed this text.*

## Reviewer 3 (Anonymous)
**Basic reporting**
.
**Experimental design**
.
**Validity of the findings**
.
**Comments for the author**
General Comments

This paper makes natural history observations about the use of salmon resources by terrestrial consumers. The observations, though relatively basic, are valuable to our understanding of the direct effects of salmon subsidies. I look forward to seeing this work published and I hope to see more basic natural history work given some of the holes in our understanding of salmon subsides.

Since this paper is putatively about vertebrate consumers, it would be helpful if it would make a small effort to consider the work that has been done on vertebrate consumers that lack feathers and fur. Stream ecologists have done most of the work on salmon subsidies, and they may be confused by the terminology in this manuscript and its tendency to ignore prior work in freshwater environments. For example, the list of vertebrate consumers in the intro does not include fishes! I realize that PeerJ is about getting the data out there and less about placing results in a broader context, but as long as the format is going to include a results and discussion section, I think it's reasonable to expect manuscripts to make a decent attempt at placing new material within the context of existing work. See comments below for specific examples.

*This point is well taken, but we are wildlife ecologists interested in how terrestrial wildlife respond to the availability of these fish. Similarly, researchers focusing on subsidies to stream invertebrates or fish should not be expected to reference the influence of salmon on bald eagles or bears. We now include text in the introduction and specify our terrestrial focus.*

While I agree that this paper has some management implications, the discussion does not put forward arguments that are either logically strong, or well supported by existing work. The paper makes little attempt to reference existing work that considers how fisheries can affect salmon availability across space and time (i.e. Doctor et al. 2010 Trans. Am. Fish Soc., Boatright et al. 2004 Trans. Am. Fish. Soc.) or how salmon abundance affects consumer foraging opportunities (Bentley et al. Ecosphere).

*This literature is important and is now referenced in our discussion of potential management recommendations.*

Specific Comments

12: "distinct life histories" seems like an overstatement, many salmon consumers opportunistic foragers

*Yes, but some can move great distances to capitalize on the best resources in the region year round, while others can deposit large quantities of fat and then fast/hibernate, and others can't do either*

*particularly well. The movers (like eagles) and the fat depositors (like bears) likely receive much more benefit than do small carnivores that must face seasonal food shortages, and so it's not surprising that bear and eagle densities are so dramatically higher in areas with lots of salmon, whereas no one has demonstrated that this is true for, say, coyotes.*

34: what about fishes?

*Point well taken. The previous line does refer to fish, and we now specify "Terrestrial vertebrate consumers…" to frame the focus of this paper.*

39: I didn't totally follow this sentence

*This has now been changed to "Consumers with distinct life histories might also use salmon differently depending on the phenology of the run and the spatial dispersion of the salmon resource."*

48: Smaller birds have higher caloric demands per unit body mass

*True, but certainly on a per-capita basis eagles require more salmon than do smaller birds.*

51: the phrase distinct life-histories is used a few times but I don't get much meaning out of it

*We go into detail describing these distinct life histories with the bear and eagle examples in the subsequent paragraph.*

58: Don't eagles also deposit fat (though not to the same degree) and don't bears also track asynchronous run-timings? Lisi et al. 2012 Geomorphology showed that substantial phenological variation in salmon can be expressed over small spatial extents, within the range of consumers with modest mobility. In fact due to limited spatial autocorrelation in salmon phenology, the relationship between spatial extent and salmon phenological extent may asymptote within a bear's home range.

*Life history strategies are always a matter of scale. Bears can move a few watersheds, but eagles can (and do) move from Northern Southeast Alaska down to Admiralty Island and then down to British Columbia in short order. Similarly, eagles face energetic hardships in winter and must rely on lipid reserves, but this is not comparable to the approximately half year hibernation of bears. Regional salmon availability influences where eagles forage, which is quite distinct from how salmon influences bears.*

64: I've seen >70 bears on a 3 km stream with the last run of salmon in a watershed, I'm not sure that late season aggregations are unique to eagles.

*Again, this is true at the watershed scale, although we have our doubts about how ubiquitous this phenomenon is because the GPS collared bears (granted we only have data from two) at Chilkoot Lake do not go to feed on the much later Chum salmon runs on the Chilkat (See Fig. 1). It's a short Euclidean distance, but there is a mountain in the way. Perhaps the less rugged topography of Bristol Bay makes it more energetically profitable for bears to move among salmon runs. In contrast, thousands of eagles can go to the latest salmon runs in all of Southeast Alaska. We have no*

*compelling evidence that bears do this to anywhere near the extent that eagles do. You may be interested to see our tracks of eagles GPS tagged on the Chilkat at:*
*http://www.ecologyalaska.com/eagle-tracker/*

69: the effects of stream size on bear predation is well documented by Quinn, Carlson, Hendre and others and could be cited here.
69: Habitat features have been shown to strongly influence whether fishes can exploit salmon subsidies: Armstrong et al. 2010 Ecology Thermal heterogeneity mediates the effects of pulsed subsidies across a landscape…

*Quinn et al. 2001 is a key paper relevant to habitat features and bears that we now cite here.*

83: What is a pool off of a larger river? Floodplain habitat?

*Sockeye in the Chilkat watershed spawn in both Chilkat Lake and in pools of slow moving water off of the mainstem of the river. These pools are typically shallow, spring fed, and clear unlike the very silty Chilkat River. Photos can be seen in Figure 2A-B. Here and elsewhere we now reference the stream ecology terminology "alcove" to describe these habitats.*

89: a sentence reiterating the methodologies would be helpful, so that we know what monitoring an individual carcass means.

*The previous paragraph reads:*
"Here we report the findings of a study in which we used remotely-triggered camera traps to quantify how wildlife foraging activity varies at salmon spawning grounds (Shardlow and Hyatt 2013), and rates of scavenging on individual salmon carcasses deposited on the forest floor by brown bears."

*Further details are later described in the methods section.*

93: I'd never heard of lake-spawning coho salmon, are you sure they are spawning along the lake sure and not simply aggregating there? Interesting regardless.

*Yes, coho can spawn on lake shores. Chilkat Lake coho spawning has been formally studied with telemetry (Ericksen and Chappel 2005, Production and Spawning Distribution of Coho Salmon from the Chilkat River, 2002-2003). Chilkoot Lake spawning location information comes from an Alaska Department of Fish and Game biologist who also did his MS on the Chilkoot River (Anthony Crupi), and from the Alaska State Parks interpretive sign at the Chilkoot River that shows the northern lakeshore as spawning habitat. We deem these sources extremely reliable, but we did not monitor coho spawning grounds as part of this research.*

96: How were the salmon breeding phenologies determined?

*The breeding phenologies were determined by direct observation and repeated visits to spawning areas over the last four years.*

114: Not a big deal, but usually freshwater scientists don't call small bodies of flowing water "creeks",

they call them "streams" and only use creek in the name.

*Yes, creek is more colloquial and connotes small size, and of course these streams all have "Creek" in the name.*

116: It'd be nice if you could define what you mean by a pool in terms that would make sense to a stream ecologist – the current definition is likely not what stream ecologists think of as a pool. The results of this paper are relevant to freshwater folks studying salmon subsidies, so to maximize the impact it would help to use clear terminology.

*After consultation with stream ecologists, we have now added the terminology "alcove" to the text.*

139: What was the condition of the carcass or how were they aquired? Since the energy density of spawning salmon declines over time (Hendry and Berg 1999 Can. J. Zool) the state of the carcass could influence its appeal to scavengers.

*Carcasses were locally collected to determine which species consume carcasses when they are available as opposed to an experiment where we deposited fresh salmon or salmon at another time of year. We now specify that:*

"Each camera was baited with a single salmon carcass staked to the ground. Salmon carcasses were collected from the adjacent spawning grounds and were monitoring during the period of the salmon run."

146: I'm not sure I find the total aggregate # of images useful, but since it's listed I'm curious of whether false triggers are included and whether the cameras worked continuously and were never taken out by bears or ran out of batteries.

*We now only include images that were tagged with an animal in the frame.*

e.g., "We obtained over 35000 images of animals from"

*Batteries miraculously didn't die and bears didn't destroy cameras (steel lock boxes), but we lost some data in 2011 due to a low quality brand of SD card.*

154: Quinn et al. 2001 do not suggest that bears do not feed along large rivers or lake shores, but instead that they exert lower predation rates on salmon in these habitats. I don't think that the observation that bears still show up at rivers and lakes shores will be surprising to anyone, and it has also been documented in the literature. I'm not saying that this documentation is not a worthy contribution for natural history, just that it shouldn't be pitched as contrary to the results of Quinn et al. 2001 at least not without more nuance.

*We have removed this reference and sentence.*

170: The tricky part about making inferences from these data is that we don't know all the salmon foraging habitats that were within the foraging neighborhoods of these consumers—if the birds didn't show up at a small stream with a camera on it, was that because they don't like small streams, or because they were on another small stream where a camera was not present? I'm not saying that these

data don't provide insights, but if the authors could address this potential issue it would make their arguments stronger.

*This paragraph has been changed in response to other reviewer comments.*

185: If brown bears are quickly grabbing many of the carcasses, how does that affect the inference of scavenging by other consumers? Would you see potentially different results if you surrounded the carcasses with electric fences permeable to small carnivores?

*Bears visited carcasses throughout the study, but did not typically feed on them until the end of the late chum salmon run.*

"Visitation of wildlife to salmon carcasses occurred consistently at each site, but consumption of carcasses was observed less frequently. While small carnivores such as mink and marten and avian scavengers were observed feeding on individual carcasses at all sites throughout the duration of the carcass monitoring, bears rarely fed on carcasses until the end of the late chum salmon run at Herman Creek."

192: What would prevent eagles from foraging in forested habitats? I'm just curious since they seem invulnerable to predation from all but bears, but they do seem to prefer open habitats.

*We are also curious about this. We can only hypothesize that they are not built for flying through forests.*

197: I've noticed that the time of year strongly influences whether bears will consume carcasses left as bait. In unpublished studies of maggot scavenging dynamics, bears would grab a carcass within a couple hours at the very beginning of the salmon runs. In contrast (and intuitively) during the peak of the runs, most carcasses left out were not scavenged.

*We have noticed similar temporal dependence, and that time of year matters because by mid-September invertebrate activity has slowed and carcasses remain relatively fresh for longer. Late in the year, carcasses not scavenged by bears during the peak of the run may get scavenged later.*

203: I am skeptical that managers would get much traction by arguing that a recent study showed bears fed on a variety of salmon runs, and therefore fishing levels should decline. The argument for backing off of chum and pink runs is interesting and perhaps ecologically valid, but those aren't necessarily "low value" species. They may be on a per pound basis, but certainly the aggregate value of pink and chum fisheries can be substantial. Further, many Alaskans (whether native, Caucasian, "urban", and rural) have little interest in maximizing brown bear productivity, but are very concerned about salmon yields.

*We have added text about the relative values of these fish in response to other reviewer comments. We agree that many Alaskans are not interested in increased bear productivity, but this is not the case everywhere. Locally, bear guides, ecotourism operators, conservationists, and wildlife viewers care. Certainly managers in Kodiak care and are conducting related research. But most importantly, this matters for currently struggling bear populations where salmon are not as plentiful such as interior British Columbia where bears are provincially threatened.*

213: Could the absence of consumers on early runs be related to other factors, such as constraints due to reproduction, or use of alternative food sources? For example some herbivores catch the "green

wave" late and leave it early, presumably due to trade-offs between resource tracking and arrival at breeding sites (Kolzsch et al. 2015 J Anim. Ecol.)

*The consumers that were absent from early pink salmon runs, i.e. gulls and eagles, were extremely abundant at the midseason lower Chilkoot River pink run (We have counted thousands of gulls and ~60 eagles on this stretch of river), which occurs over a wide open area rather than a forested stream). We did not monitor this run with cameras because it is heavily used by humans. We now state this in the Discussion.*

228: it's not clear whether this is speculation or backed by existing data or studies

*Changed to clarify that this is speculation based on natural history observations.*

234: the term integrate is confusing when applied to energy allocation

*Changed to "effectively exploit"*

242: Seems worth noting that salmon carcasses are likely much lower in energy density than the alternative prey of small carnivores—a spawned out salmon is about ~3 kJ/g (Hendry and Berg 1999) whereas a rodent is about 2-3-times as energy dense (Cox and Secor 2007, Comp. Biochem. Phys A). Foraging theory suggests that the abundance of preferred prey (not alternative prey) influences prey switching, which in turn suggests that small carnivores might only switch to salmon when rodent cycles are in their low phase. See Lisi et al. 2013 Ecology of Freshwater Fish for twist on fish switching to rodent prey during years with low salmon abundance.

*This is an excellent point, and a hypothesis worth testing. We now cite these papers and describe the possibility that small carnivores may prefer small mammal prey and switch to salmon when small mammals are at low abundance. The consumption of shrews shown in Lisi et al. is amazing.*

248: Combining all salmon biomass when managing for salmon misses the point that certain salmon populations have disproportionate importance to consumers, which I thought was a key point of this paper. I'm not saying it's feasible to manage salmon fisheries for specific populations, but it would certainly seem important to avoid management practices that are likely to hammer specific populations, for example the common practice of meeting escapement goals and then heavily harvesting late arriving fish.

*We agree that this was not nuanced enough. We have changed the text to:*

"Although there are quantitative differences in the nutritional quality of salmon and their accessibility, all salmon on the landscape were heavily used by bears.  In contrast, the activity pattern of bald eagles suggests that fisheries management that considers their nutritional requirements might instead focus on increasing escapements at late chum salmon runs where eagles congregate in Southeast Alaska, British Columbia, and Northern Washington."

255: these results seem overstated, I would say "would likely" but "will" is not demonstrated by these data.

*Agreed. Changed.*

Figure 1: Can you color or symbol the sites by run timing and or species

*Species are now colored and labeled on the map, and we have changed the figure legend to describe the run timings.*

Levin, P. S., M. J. Fogarty, S. A. Murawski, and D. Fluharty. 2009. Integrated ecosystem assessments: developing the scientific basis for ecosystem-based management of the ocean. PLOS Biology **7**:23-28.

Schindler, D. E., J. B. Armstrong, K. T. Bentley, K. Jankowski, P. J. Lisi, and L. X. Payne. 2013. Riding the crimson tide: mobile terrestrial consumers track phenological variation in spawning of an anadromous fish. Biology Letters **9**:1-4.

Shardlow, T. F., and K. D. Hyatt. 2013. Quantifying associations of large vertebrates with salmon in riparian areas of British Columbia streams by means of camera-traps, bait stations, and hair samples. Ecological Indicators **27**:97-107.